# OpenReview forum: "Latent Thoughts Tuning: Bridging Context and Reasoning with Fused Information in Latent Tokens"
_ICML.cc/2026/Conference — ICML 2026 regular_

### Official Review · Reviewer_7RAf · 2026-03-03

**Soundness:** 3
**Presentation:** 3
**Significance:** 3
**Originality:** 3
**Overall Recommendation:** 5
**Confidence:** 4

**Summary:**

This paper introduces LT-Tuning, a training paradigm for latent reasoning models, addressing the instability issue of latent reasoning models. LT-Tuning uses a confidence-driven reasoning strategy to adaptively switch reasoning manner.

**Compliance With Llm Reviewing Policy:**

Affirmed.

**Final Justification:**

My concerns are addressed. This is a good paper and I'll raise my score from borderline accept to accept.

**Key Questions For Authors:**

Please refer to weakness.

**Limitations:**

1. Only evaluated on simple benchmarks (grade-school mathematics reasoning)

**Strengths And Weaknesses:**

Strength:
1. The overall performance of LT-Tuning is impressive: it outperforms both latent reasoning baslines, and also the explicit CoT method.
2. This paper is well written and easy to follow.


Weakness:
1. Where are the efficiency results? This paper claims efficiency but it seems that there's not an experiment supporting this.
2. I suggest adding discussion on SwiReasoning (ICLR'26), which also proposes confidence-driven reasoning manner switching.

---

> ### Author Rebuttal · Authors · 2026-03-31
>
> We sincerely thank the reviewer for the positive assessment and the concrete suggestions.
>
> **[W1] Efficiency results.**
>
> We provide a direct token-efficiency analysis on Llama-3.1-8B:
>
> | Dataset | CoT Acc (%) | LT Acc (%) **(ours)** | CoT Tok | LT Tok | Reduction |
> |---|---:|---:|---:|---:|---:|
> | GSM8K | 49.5 | 58.1 | 289.1 | 225.1 | **22.1%** |
> | ASDiv-Aug | 69.6 | 72.2 | 148.7 | 108.8 | **26.9%** |
> | MultiArith | 78.3 | 92.8 | 195.0 | 136.8 | **29.9%** |
> | SVAMP | 49.3 | 52.3 | 185.2 | 136.3 | **26.4%** |
> | **Overall** | **61.7** | **68.8** | **220.7** | **167.5** | **24.1%** |
>
> LT-Tuning achieves +7.1% accuracy with 24.1% fewer text tokens. We will revise the paper to frame efficiency as reduced reliance on explicit CoT tokens.
>
> **[W2] Discussion of SwiReasoning (ICLR'26).**
>
> Thank you for this pointer. The two methods are distinct and complementary:
>
> 1. **Setting:** SwiReasoning is **training-free** and targets already-strong reasoning LLMs (e.g., Qwen3-8B). LT-Tuning is a **post-training** framework that endows base models with latent reasoning capabilities.
> 2. **Granularity:** SwiReasoning switches at the **thinking-block level** via entropy trends. LT-Tuning operates at **token-level** via prediction confidence.
> 3. **Scope:** SwiReasoning focuses on *when* to switch modes. LT-Tuning additionally addresses *how* to construct stable latent tokens via Context-Prediction Fusion—shown to be critical for preventing feature collapse at scale (Table 2: −23.5% without it on 8B).
>
> We will add this comparison in the revision.
>
> **[L1] Only evaluated on simple benchmarks.**
>
> We added two evaluations beyond the original setup:
>
> *(a) Cross-family (Qwen2.5):*
>
> *Given the limited rebuttal window, we currently report Qwen2.5 results at the 0.5B and 1.5B scales, and will further extend the study to larger models in the revision.*
>
> | Model | GSM8K (%) | ASDiv (%) | MultiArith (%) | SVAMP (%) | Avg (%) |
> |---|---:|---:|---:|---:|---:|
> | Qwen2.5-0.5B CoT | 30.3 | 60.6 | 81.7 | 36.7 | 52.3 |
> | Qwen2.5-0.5B SoftThinking | 28.6 | 59.3 | 83.3 | 35.7 | 51.7 |
> | Qwen2.5-0.5B LT-Tuning | 31.1 | 65.1 | 80.1 | 38.3 | **53.6(+1.3)** |
> | Qwen2.5-1.5B CoT | 45.3 | 62.2 | 93.9 | 48.0 | 62.3 |
> | Qwen2.5-1.5B SoftThinking | 46.1 | 66.8 | 88.7 | 49.0 | 62.7 |
> | Qwen2.5-1.5B LT-Tuning | 47.7 | 70.9 | 93.9 | 48.3 | **65.2(+2.5)** |
>
> *(b) Non-math (GPQA-Diamond, trained on Bespoke-Stratos-17k):*
>
> | Model | GPQA-Diamond Acc.(%) |
> |---|---:|
> | Llama-3.2-1B CoT | 25.7 |
> | Llama-3.2-1B SoftThinking | 21.2 |
> | Llama-3.2-1B LT-Tuning | **27.8(+2.1)** |
>
> LT-Tuning generalizes across model families and to non-math reasoning.

---

> > ### Author Rebuttal · Reviewer_7RAf · 2026-04-02
> >
> > Thank you for your detailed response. Most of my concerns are addressed. I will raise my score accordingly.
> >
> > Have to say, I completely understand the newly added experiments are conducted on relatively smaller models, given that there's only several days for the authods to prepare for experimental setup and model training. I wish that the authors could apply LTT on larger models and more mordern benchmarks in the final revision.

---

> > > ### Author Response · Authors · 2026-04-06
> > >
> > > Thank you very much for the positive follow-up and for raising your score. We are glad that the added efficiency analysis and discussion addressed your concerns. We also appreciate your suggestion on larger models and more modern benchmarks, and we will extend the study in these directions in the revision.

---

### Official Review · Reviewer_NuhK · 2026-03-08

**Soundness:** 2
**Presentation:** 2
**Significance:** 2
**Originality:** 2
**Overall Recommendation:** 3
**Confidence:** 3

**Summary:**

This paper is about latent reasoning for LLMs.
Instead of forcing the model to write every intermediate reasoning step as human-readable text, the proposed method wants the model to think in continuous latent states.
The motivation is straightforward: explicit CoT can be helpful, but it is slow, verbose, and constrained by natural-language tokens.
The paper’s central claim is that better latent-token construction can make latent reasoning both more stable and more effective compared with other baseline methods.
The proposed LT-Tuning has three stages.
First, the model is fine-tuned with ordinary explicit CoT.
Second, the training data is modified so that when the model is uncertain about the next token, the sequence inserts a special placeholder to let the model know when to use a latent step instead of directly emitting text.
Third, the latent token is built by fusing the previous hidden state and a probability-weighted embedding of likely next tokens.
Empirical results show consistent performance gains on math reasoning benchmarks with the LLaMA series.

**Compliance With Llm Reviewing Policy:**

Affirmed.

**Final Justification:**

Based on the rebuttal, the method may be a good and practically useful method, but I still feel the current manuscript is not yet ready since it's lack of empirical results, theoretical analysis, and novelty. So I maintain my score.

**Key Questions For Authors:**

1. What is the essential difference between the proposed latent-token method and other related methods? Is it only combining hidden states and embeddings, or something else I'm missing?

2. Can the authors provide a more principled theoretical framework for the method?

3. Can the authors substantiate the efficiency claim more directly? Since the method is mainly motivated by reducing reliance on explicit CoT, it would strengthen the paper to report latency, token count, wall-clock, or FLOPs comparisons against explicit CoT and other latent baselines.

4. Why is the empirical validation restricted to math benchmarks and LLaMA backbones only?

**Limitations:**

Yes.

**Strengths And Weaknesses:**

Strengths

1. The paper is easy to follow. The problem setup is clear, and the three-stage training pipeline is well structured.

2. The proposed method achieves the best reported average accuracy at all three model scales, and the gains are especially notable in the 8B setting.

3. The entropy/attention analysis and PCA visualizations provide some support for the paper’s central claims.

Weaknesses

1. The main distinction from prior intrinsic latent-token methods seems to be the combination of two already-motivated ingredients. This is a sensible design, but it still feels more like a technical composition of existing ideas than a major conceptual advance.

2. There is no real theoretical framework explaining when fusion should help. As a result, the paper currently reads more as an empirical report than a work that substantially deepens the understanding of latent reasoning.

3. All experiments are on math benchmarks, with all models trained on GSM8K and tested on four related arithmetic datasets. In addition, only Llama-family backbones are used.

4. The paper claims that it is motivated through reduced reliance on verbose text reasoning and improved efficiency, but I did not see convincing runtime-oriented evidence such as latency, FLOPs, token cost, or wall-clock comparisons.

---

> ### Author Rebuttal · Authors · 2026-03-31
>
> We sincerely thank the reviewer for the constructive feedback and for recognizing the strong performance and supportive analyses.
>
> **[Q1] Essential difference from prior methods?**
>
> We believe the contribution goes beyond combining two vectors. Our central finding is that **distribution mismatch is the scaling bottleneck** for intrinsic latent reasoning. The evidence is a striking asymmetry in our ablation (Table 2): on 8B, removing fusion (w/o Stage 3) causes −23.5%, while removing latent reasoning entirely (w/o Latent) causes only −7.2%. This means **poorly constructed latent tokens are worse than no latent tokens at all**.
>
> LT-Tuning is a unified response to this finding: (1) Context-Prediction Fusion controls latent-token quality, (2) confidence-driven insertion controls when latent reasoning is used, and (3) the progressive curriculum makes this optimization stable. The ablation confirms each is necessary.
>
> **[Q2] Theoretical framework.**
>
> We offer a principled justification that directly connects to our empirical findings.
>
> **(i) Bayes-Optimality.** The predictive embedding $e_\mathrm{pred}=\sum_w p(w|c)E(w)$ is the unique MSE-optimal estimate of the next-token embedding under the model's own distribution. This grounds its use as a fusion component rather than an ad hoc heuristic.
>
> **(ii) Mismatch Contraction.** Let $\mathcal{M}=\mathrm{conv}\{E(w):w\in\mathcal{V}\}$ be the convex hull of token embeddings. Since $e_\mathrm{pred}\in\mathcal{M}$ by construction, our fusion $z_\alpha=\alpha h+(1-\alpha)e_\mathrm{pred}$ satisfies:
>
> $$\mathrm{dist}(z_\alpha,\mathcal{M}) \le \alpha\cdot\mathrm{dist}(h,\mathcal{M})$$
>
> Whenever $h\notin\mathcal{M}$, any $\alpha<1$ **strictly contracts** the off-manifold distance versus directly reusing $h$ (Coconut). This formalizes the mechanism behind feature-collapse mitigation.
>
> **(iii) Mismatch-Aware Optimal Fusion.** Modeling $h$ and $e_\mathrm{pred}$ as complementary noisy observations of a reference latent representation and adding a manifold-compatibility penalty $\lambda\,\mathrm{dist}(z_\alpha,\mathcal{M})^2$, the optimal weight becomes:
>
> $$\alpha^\dagger = \frac{\sigma_e^2-\rho}{\sigma_h^2+\sigma_e^2-2\rho+\lambda D_h^2}$$
>
> When the two views carry non-redundant information ($\rho<\min(\sigma_h^2,\sigma_e^2)$), $\alpha^\dagger$ lies strictly in $(0,1)$. Our ablation confirms this, as removing either component degrades accuracy. Moreover, the formula yields a testable prediction: **larger mismatch $D_h$ → smaller $\alpha^\dagger$ → more reliance on the predictive component**. This matches our results exactly: removing fusion hurts 8B far more (−23.5%, untied embeddings, large $D_h$) than 3B (−8.0%, tied embeddings, small $D_h$). Full proofs will be provided in the revision.
>
> **[Q3] Efficiency evidence.**
>
> We measured the average number of generated text tokens on Llama-3.1-8B to directly quantify the reasoning compression effect:
>
> | Dataset | CoT Acc  (%) | LT Acc  (%) (ours) | CoT Tok | LT Tok | Reduction |
> |---|---:|---:|---:|---:|---:|
> | GSM8K | 49.5 | 58.1 | 289.1 | 225.1 | **22.1%** |
> | ASDiv-Aug | 69.6 | 72.2 | 148.7 | 108.8 | **26.9%** |
> | MultiArith | 78.3 | 92.8 | 195.0 | 136.8 | **29.9%** |
> | SVAMP | 49.3 | 52.3 | 185.2 | 136.3 | **26.4%** |
> | **Overall** | **61.7** | **68.8** | **220.7** | **167.5** | **24.1%** |
>
> LT-Tuning achieves +7.1% accuracy while generating 24.1% fewer text tokens, confirming that latent tokens effectively compress verbose intermediate reasoning. We will revise the paper to frame efficiency as reduced reliance on explicit CoT.
>
> **[Q4] Why only math + LLaMA?**
>
> We added two new settings:
>
> *(a) Cross-family generalization (Qwen2.5):*
>
> *Given the limited rebuttal window, we currently report Qwen2.5 results at the 0.5B and 1.5B scales, and will further extend the study to larger models in the revision.*
>
> | Model | GSM8K (%) | ASDiv (%) | MultiArith (%) | SVAMP (%) | Avg (%) |
> |---|---:|---:|---:|---:|---:|
> | Qwen2.5-0.5B CoT | 30.3 | 60.6 | 81.7 | 36.7 | 52.3 |
> | Qwen2.5-0.5B SoftThinking | 28.6 | 59.3 | 83.3 | 35.7 | 51.7 |
> | Qwen2.5-0.5B LT-Tuning | 31.1 | 65.1 | 80.1 | 38.3 | **53.6(+1.3)** |
> | Qwen2.5-1.5B CoT | 45.3 | 62.2 | 93.9 | 48.0 | 62.3 |
> | Qwen2.5-1.5B SoftThinking | 46.1 | 66.8 | 88.7 | 49.0 | 62.7 |
> | Qwen2.5-1.5B LT-Tuning | 47.7 | 70.9 | 93.9 | 48.3 | **65.2(+2.5)** |
>
> *(b) Non-math generalization (GPQA-Diamond, trained on Bespoke-Stratos-17k):*
>
> | Model | GPQA-Diamond Acc.(%) |
> |---|---:|
> | Llama-3.2-1B CoT | 25.7 |
> | Llama-3.2-1B SoftThinking | 21.2 |
> | Llama-3.2-1B LT-Tuning | **27.8(+2.1)** |
>
> The Qwen results show cross-family transfer, and GPQA (trained on Bespoke-Stratos-17k) confirms the framework is not math-specific. We will incorporate broader evaluations in the revision.

---

> > ### Author Rebuttal · Reviewer_NuhK · 2026-04-02
> >
> > Thank you for the detailed rebuttal. The response is helpful and does strengthen the paper in several ways. However, I believe my concerns are partially addressed, not fully resolved.
> >
> > In particular, while the added Qwen and GPQA results are useful, the performance gains are still relatively modest, so I have to doubt about the robustness on the generalization beyond the original setting (math/LLaMA). Similarly, the added theory part is promising, but it is currently still more of a sketch than a fully established framework. Moreover, on efficiency, the response only reports generated tokens, not latency, FLOPs, wall-clock, KV-cache cost, or broader baseline comparisons. The methodological design also seems to be engineered combination, and the novelty still appears moderate in its current form.
> >
> > Overall, I agree that this may be a good and practically useful method, but I still feel the current manuscript is not yet complete enough. The work would benefit from another round of revision before it can be more confidently presented.

---

> > > ### Author Response · Authors · 2026-04-06
> > >
> > > Thank you for the careful follow-up. We appreciate the reviewer’s balanced assessment and the recognition that the method is practically useful. In the revision, we will provide a complete theoretical analysis with full proofs, incorporate the added Qwen and GPQA experiments into the final version, and continue extending the study to larger models and broader tasks to further examine the scaling and robustness issues. We thank the reviewer again for highlighting these important directions, which will directly help strengthen the final version.

---

### Official Review · Reviewer_wBst · 2026-03-12

**Soundness:** 2
**Presentation:** 3
**Significance:** 2
**Originality:** 2
**Overall Recommendation:** 3
**Confidence:** 4

**Summary:**

The authors propose LT-Tuning where the authors first look for token positions with low confidence and replace them with thinking tokens whose embedding is a weighted sum of original hidden states and predictive soft embedding derived from the model's output distribution. Experiments on mathematical reasoning benchmarks show that proposed LT-Tuning consistently improve performance across Lama.3.2-1B, 3B, 8B.

**Compliance With Llm Reviewing Policy:**

Affirmed.

**Final Justification:**

The authors have addressed cross-model generalization and extended the evaluation to other domains such as GPQA. Nevertheless, it remains uncertain whether the proposed method scales effectively to more advanced models and more challenging datasets. I believe the current manuscript would be strengthened by an additional round of experiments that could more thoroughly examine scalability. Accordingly, I've updated the score 2 -> 3.

**Key Questions For Authors:**

See Weaknesses section. If the authors could directly address these points, it would substantially strengthen the paper and help resolve any remaining concerns / misunderstandings.

**Limitations:**

impact statement is well written. For limitations, i would refer to the weaknesses discussed above.

**Strengths And Weaknesses:**

Strength

S1. The paper is clearly written and easy to follow.

S2. The paper well identifies important bottlenecks in latent reasoning, which lead to conceptually interesting method.

S3. The paper includes good diagnostic analysis.

Weaknesses

W1. Evaluation scope is somewhat narrow.

(1) While the authors used MetaMath (Yu, 2023) for training, there is no performance reported on MATH500.

(2) It is yet unclear how the method generalizes to other model families, such as Qwen.

(3) Also, it is unclear as to whether the proposed method can generalize to non-math domain, i.e. GPQA, MBPP, etc.

W2. While the paper provides empirical intuition for why the fused embedding is effective, the justification is somewhat heuristic. The method would be strengthened if a more principled or theoretical explanation is provided.

W3. The paper seems to be missing compute-efficiency analysis compared to its baselines.

---

> ### Author Rebuttal · Authors · 2026-03-31
>
> We thank the reviewer for recognizing the clear writing and good diagnostic analysis. We address each concern with new evidence.
>
> **[W1] Evaluation scope.**
>
> We agree that broader evaluation would strengthen the paper. We would also like to gently clarify one factual point first: our math experiments are trained on **GSM8K**, not MetaMath. We added two complementary extensions that directly address the concerns about model-family transfer and non-math generalization.
>
> *(a) Cross-family generalization (Qwen2.5):*
>
> *Given the limited rebuttal window, we currently report Qwen2.5 results at the 0.5B and 1.5B scales, and will further extend the study to larger models in the revision.*
>
> | Model | GSM8K (%) | ASDiv (%) | MultiArith (%) | SVAMP (%) | Avg (%) |
> |---|---:|---:|---:|---:|---:|
> | Qwen2.5-0.5B CoT | 30.3 | 60.6 | 81.7 | 36.7 | 52.3 |
> | Qwen2.5-0.5B SoftThinking | 28.6 | 59.3 | 83.3 | 35.7 | 51.7 |
> | Qwen2.5-0.5B LT-Tuning | 31.1 | 65.1 | 80.1 | 38.3 | **53.6(+1.3)** |
> | Qwen2.5-1.5B CoT | 45.3 | 62.2 | 93.9 | 48.0 | 62.3 |
> | Qwen2.5-1.5B SoftThinking | 46.1 | 66.8 | 88.7 | 49.0 | 62.7 |
> | Qwen2.5-1.5B LT-Tuning | 47.7 | 70.9 | 93.9 | 48.3 | **65.2(+2.5)** |
>
> *(b) Non-math generalization (GPQA-Diamond, trained on Bespoke-Stratos-17k):*
>
> | Model | GPQA-Diamond Acc.(%) |
> |---|---:|
> | Llama-3.2-1B CoT | 25.7 |
> | Llama-3.2-1B SoftThinking | 21.2 |
> | Llama-3.2-1B LT-Tuning | **27.8(+2.1)** |
>
> **[W2] Theoretical justification for fusion.**
>
> We provide a principled analysis resting on three formal results.
>
> **(i) Bayes-Optimality.** The predictive embedding $e_\mathrm{pred}=\sum_w p(w|c)E(w)$ is the unique MSE-optimal estimate of the next-token embedding under the model's own distribution. This grounds its use as a fusion component rather than an ad hoc heuristic.
>
> **(ii) Mismatch Contraction.** Let $\mathcal{M}=\mathrm{conv}\{E(w):w\in\mathcal{V}\}$ be the convex hull of token embeddings. Since $e_\mathrm{pred}\in\mathcal{M}$ by construction, our fusion $z_\alpha=\alpha h+(1-\alpha)e_\mathrm{pred}$ satisfies:
>
> $$\mathrm{dist}(z_\alpha,\mathcal{M}) \le \alpha\cdot\mathrm{dist}(h,\mathcal{M})$$
>
> Whenever $h\notin\mathcal{M}$, any $\alpha<1$ **strictly contracts** the off-manifold distance versus directly reusing $h$ (Coconut). This formalizes the mechanism behind feature-collapse mitigation.
>
> **(iii) Mismatch-Aware Optimal Fusion.** Modeling $h$ and $e_\mathrm{pred}$ as complementary noisy observations of a reference latent representation and adding a manifold-compatibility penalty $\lambda\,\mathrm{dist}(z_\alpha,\mathcal{M})^2$, the optimal weight becomes:
>
> $$\alpha^\dagger = \frac{\sigma_e^2-\rho}{\sigma_h^2+\sigma_e^2-2\rho+\lambda D_h^2}$$
>
> When the two views carry non-redundant information ($\rho<\min(\sigma_h^2,\sigma_e^2)$), $\alpha^\dagger$ lies strictly in $(0,1)$. Our ablation confirms this, as removing either component degrades accuracy. Moreover, the formula yields a testable prediction: **larger mismatch $D_h$ → smaller $\alpha^\dagger$ → more reliance on the predictive component**. This matches our results exactly: removing fusion hurts 8B far more (−23.5%, untied embeddings, large $D_h$) than 3B (−8.0%, tied embeddings, small $D_h$). Full proofs will be provided in the revision.
>
> **[W3] Compute-efficiency analysis.**
>
> | Dataset | CoT Acc (%) | LT Acc (%) **(ours)** | CoT Tok | LT Tok | Reduction |
> |---|---:|---:|---:|---:|---:|
> | GSM8K | 49.5 | 58.1 | 289.1 | 225.1 | **22.1%** |
> | ASDiv-Aug | 69.6 | 72.2 | 148.7 | 108.8 | **26.9%** |
> | MultiArith | 78.3 | 92.8 | 195.0 | 136.8 | **29.9%** |
> | SVAMP | 49.3 | 52.3 | 185.2 | 136.3 | **26.4%** |
> | **Overall** | **61.7** | **68.8** | **220.7** | **167.5** | **24.1%** |
>
> LT-Tuning achieves +7.1% accuracy while generating 24.1% fewer text tokens, confirming a favorable accuracy–efficiency tradeoff.

---

> > ### Author Rebuttal · Reviewer_wBst · 2026-04-02
> >
> > Dear Authors,
> >
> > Thank you for the detailed response, and I apologize for my earlier confusion regarding Metamath.
> >
> > While some of my concerns have been addressed, a few still remain. Given the limited rebuttal window, the authors conducted experiments mainly on the 0.5B and 1.5B scales, which still do not fully establish scalability to larger and more advanced models. I also believe the paper would be further strengthened by including results on more challenging datasets in the math domain, as the current benchmarks appear relatively easy given the capabilities of recent open-source models.
> >
> > I appreciate the authors’ detailed clarifications and have accordingly updated my score.

---

> > > ### Author Response · Authors · 2026-04-06
> > >
> > > Thank you for the thoughtful follow-up, and we appreciate your updated assessment. We agree that evaluation on larger and more advanced models, as well as on more challenging math benchmarks, would further strengthen the paper. Our added Qwen and GPQA results are intended as complementary evidence that LT-Tuning is not tied to the original setting, and we will continue to extend the study to larger scales and harder math tasks in the revision.

---

### Official Review · Reviewer_ZVwi · 2026-03-12

**Soundness:** 4
**Presentation:** 4
**Significance:** 4
**Originality:** 4
**Overall Recommendation:** 5
**Confidence:** 4

**Summary:**

The paper introduces LT-Tuning (latent thoughts tuning), a novel framework for latent space reasoning. The proposed method uses a confidence-driven decision rule to determine whether to use explicit CoT or latent thoughts. They also use context-prediction fusion to construct well-aligned latent token embeddings, and use a progressive curriculum learning to avoid unstable optimization in the latent space or feature collapse. The authors conducted comprehensive experiments to demonstrate the effectiveness and scalability of LT-Tuning.

**Compliance With Llm Reviewing Policy:**

Affirmed.

**Final Justification:**

The rebuttal addressed my concerns, and I will keep my positive score.

**Key Questions For Authors:**

1. I feel confused about Equation (3) in Line 153. From my understanding of latent space reasoning (such as coconut), it should be $h\_t = F\_{\theta}(z\_1, z\_2, \ldots, z\_t)$? Or does $h\_t$ denote a sequence of continuous thoughts instead of a single step thought?
How to determine how long to train at each stage?

2. In Figure 3, it seems that when model size increases, the number of latent tokens first decreases (1B->3B), then increases (3B->8B). Do you have any guesses or an intuitive explanation for this phenomenon?

3. In Section 6, the authors show that coconut exhibits severe feature collapse. On the other hand, previous work [1] shows that the advantage of coconut compared to discrete CoT is that it can reason by maintaining a superposition of different possible reasoning trajectories, especially in tasks with large branching factors such as graph reachability. I’m curious about whether coconut also shows severe feature collapse compared to LT-Tuning in high-branching-factor tasks such as graph reachability.


**References**:

[1] Zhu, Hanlin, Shibo Hao, Zhiting Hu, Jiantao Jiao, Stuart Russell, and Yuandong Tian. "Reasoning by superposition: A theoretical perspective on chain of continuous thought." arXiv preprint arXiv:2505.12514 (2025).

**Limitations:**

Yes

**Strengths And Weaknesses:**

**Strengths:**


1. Latent space reasoning is a promising paradigm for reasoning and test-time scaling. The authors proposed a novel method that improves over previous work, such as coconut, soft-thinking, etc. The experiments are comprehensive and look solid to me.

2. The authors conducted careful ablation studies to verify the effectiveness and necessity of each component/stage of the proposed LT-Tuning.

3. The authors also conducted in-depth analyses of LT-Tuning, e.g., by examining generation entropy and the proportion of attention allocated to thinking tokens, etc.



**Weaknesses:**

 I don’t see any obvious weaknesses in this paper.

---

> ### Author Rebuttal · Authors · 2026-03-31
>
> We sincerely thank the reviewer for the strong support and the insightful questions.
>
> **[Q1] Clarification on Eq. (3) and stage training length.**
>
> Thank you for pointing this out. A more precise notation is $h_t = F_\theta(z_1, z_2, \ldots, z_t)$, as the Transformer conditions on the full latent prefix via the KV cache. Our simplified form was intended to highlight the recurrent nature of latent-state evolution. We will revise it for clarity.
>
> For the stage lengths, we chose them empirically based on stage-wise convergence and optimization stability. Stage 1 can converge quickly and establishes the model’s basic reasoning behavior. Stage 2 typically needs 1–2 epochs to learn confidence-driven insertion, and Stage 3 requires longer training (3–7 epochs depending on model scale) because learning a stable fused latent space is more challenging.
>
> **[Q2] Non-monotonic trend in `<thinking>` token counts across scales.**
>
> Thank you for this insightful observation. We do not interpret Figure 3 as a monotonic cross-scale law. The absolute number of generated `<thinking>` tokens is jointly affected by model capability / calibration and scale-specific latent-token settings (e.g., different $\tau$ and maximum latent budget $k$), so the raw counts are not directly comparable across model sizes.
>
> The main takeaway of Figure 3 is the within-model trend: for each model, more difficult questions tend to trigger more latent tokens.
>
> **[Q3] Potential Feature collapse on high-branching-factor tasks (graph reachability).**
>
> Thank you for the helpful pointer to Zhu et al. (2025). We agree this is a very relevant perspective. We additionally conducted a preliminary study on the graph reachability setting (ProsQA) using Llama3.2-1B. On this structured task, we did not observe obvious feature collapse for Coconut. Instead, likely due to the high training accuracy and overfitting, the latent-space geometry becomes stable after only 1–2 latent steps, with the relative positions across samples remaining largely fixed thereafter. Coconut reaches this early-stabilized pattern sooner, while LT-Tuning remains more sample-conditioned.

---

> > ### Author Rebuttal · Reviewer_ZVwi · 2026-04-02
> >
> > Thank the authors for the response. I would keep my score and remain positive on this paper.

---

> > > ### Author Response · Authors · 2026-04-06
> > >
> > > Thank you very much for the positive follow-up and for your supportive assessment. We are glad that our rebuttal addressed your concerns. We will incorporate the clarification of Eq. (3) and other additional discussions in the revision.

---

### Decision · Program_Chairs · 2026-04-30

**Decision:**

Accept (regular)

**Comment:**

This paper proposes a method for improving implicit reasoning in language models using a combination of confidence-driven switching, context–prediction fusion, and curriculum learning. Reviewers agree that the proposed method is effective empirically, with large improvements over implicit and explicit CoT baselines.

However, I have two main concerns that limit my confidence in the paper.

First, the evaluation does not cleanly isolate the benefit of implicit CoT versus explicit CoT. The proposed method is finetuned, while explicit CoT baselines are primarily prompting-based. This makes it difficult to determine whether the observed gains come from the implicit CoT formulation or simply from additional supervised training. A more controlled comparison (e.g., finetuned CoT baselines, note that a finetuned CoT method can easily get 60% strong on GSM-8K) would be necessary to support the paper's claims, and I highly recommend authors to add that comparison in the final version of the paper, as right now it seems to mislead readers that implicit CoT can even outperform explicit CoT.

Second, the empirical validation remains limited in scope and leaves open questions about scalability. While the rebuttal adds experiments on Qwen and GPQA-DIAMOND, these are conducted at relatively small scales and show weak performance and modest gains (e.g., even Qwen3 0.6B got 27.9% on GQPA-DIAMOND with prompting). It remains unclear whether the proposed multi-component framework would scale effectively to stronger models and more challenging benchmarks, as empirically simpler methods tend to scale better, and I encourage authors to further simplify their approach to scale up in followup works.

Overall, despite the two caveats mentioned above, I think implicit reasoning is worth further scientific exploration, and I understand the resource constraints for most researchers. Considering this and the opinions of reviewers, I recommend a weak accept.